# Non-invasive early detection of cancer four years before conventional diagnosis using a blood test

Xingdong Chen[1,2,3,12], Jeffrey Gole[4,12], Athurva Gore[4,12], Qiye He[5,12], Ming Lu[2,6,12], Jun Min[4], Ziyu Yuan[2], Xiaorong Yang[2,6], Yanfeng Jiang[1,2], Tiejun Zhang[7], Chen Suo[7], Xiaojie Li[5], Lei Cheng[5], Zhenhua Zhang[5], Hongyu Niu[5], Zhe Li[5], Zhen Xie[5], Han Shi[4], Xiang Zhang[8], Min Fan[9], Xiaofeng Wang[1,2], Yajun Yang[1,2], Justin Dang[4], Catie McConnell[4], Juan Zhang[2], Jiucun Wang[1,2,3], Shunzhang Yu[2,7], Weimin Ye[2,10✉], Yuan Gao[4✉], Kun Zhang [11✉], Rui Liu[4,5✉] & Li Jin[1,2,3✉]

Early detection has the potential to reduce cancer mortality, but an effective screening test must demonstrate asymptomatic cancer detection years before conventional diagnosis in a longitudinal study. In the Taizhou Longitudinal Study (TZL), 123,115 healthy subjects provided plasma samples for long-term storage and were then monitored for cancer occurrence. Here we report the preliminary results of PanSeer, a noninvasive blood test based on circulating tumor DNA methylation, on TZL plasma samples from 605 asymptomatic individuals, 191 of whom were later diagnosed with stomach, esophageal, colorectal, lung or liver cancer within four years of blood draw. We also assay plasma samples from an additional 223 cancer patients, plus 200 primary tumor and normal tissues. We show that PanSeer detects five common types of cancer in 88% (95% CI: 80–93%) of post-diagnosis patients with a specificity of 96% (95% CI: 93–98%), We also demonstrate that PanSeer detects cancer in 95% (95% CI: 89–98%) of asymptomatic individuals who were later diagnosed, though future longitudinal studies are required to confirm this result. These results demonstrate that cancer can be non-invasively detected up to four years before current standard of care.

[1] State Key Laboratory of Genetic Engineering and Collaborative Innovation Center for Genetics and Development, School of Life Sciences, Fudan University, 200438 Shanghai, China. [2] Taizhou Institute of Health Sciences, Fudan University, 225300 Taizhou, Jiangsu, China. [3] Human Phenome Institute, Fudan University, 201203 Shanghai, China. [4] Singlera Genomics Inc., La Jolla, CA 92037, USA. [5] Singlera Genomics (Shanghai) Ltd., 201203 Shanghai, China. [6] Clinical Epidemiology Unit, Qilu Hospital of Shandong University, 250012 Jinan, Shandong, China. [7] Department of Epidemiology, School of Public Health, Fudan University, 200032 Shanghai, China. [8] Taizhou Disease Control and Prevention Center, 225300 Taizhou, Jiangsu, China. [9] Taixing Disease Control and Prevention Center, 225400 Taizhou, Jiangsu, China. [10] Department of Medical Epidemiology and Biostatistics, Karolinska Institutet, 17177 Stockholm, Sweden. [11] Department of Bioengineering, University of California at San Diego, La Jolla, CA 92093, USA. [12] These authors contributed equally: Xingdong Chen, Jeffrey Gole, Athurva Gore, Qiye He, Ming Lu. ✉email: weimin.ye@ki.se; gary.gao@singleragenomics.com; kzhang@bioeng.ucsd.edu; rliu@singleragenomics.com; lijin@fudan.edu.cn

Late stage cancers often lack an effective treatment option[1,2]. Survival rates increase significantly when cancer is identified at early stages, as the tumor can be surgically removed or treated with milder drug regimens[3]; average 5-year survival at early stage is 91%, while average 5-year survival at late stage is 26%[4]. Detection of tumors at the earliest possible stage is therefore of paramount importance for cancer treatment. Currently, a limited number of screening tests exist for a few cancer types, including colonoscopy[5], prostate specific antigen[6], mammography[7], and cervical cytology[8]. However, the efficacy of some tests has been questioned[9], and many patients do not follow medical guidelines for screening[10]. Most cancer types currently lack an effective non-invasive early screening option[11]. Importantly, a formal demonstration of early detection requires collecting samples years before conventional cancer diagnosis, which is only feasible with longitudinal tracking of a large number of healthy individuals and identifying the very small fraction who develop cancer over time (at the incidence rate of cancer in the general population).

Recently, circulating tumor DNA (ctDNA) in blood plasma has become a promising cancer biomarker[12]. ctDNA has been demonstrated to have utility for non-invasive detection of cancer[13–17], personalized treatment of late stage cancer[18], and residual monitoring of cancer during and after treatment[19,20]. However, current detection studies have mostly focused on detecting cancer in patients who have already been diagnosed[21], though a few studies have shown cancer detection prior to conventional diagnosis in limited cancer types[13,14,17]. While ctDNA has the potential for early diagnosis, several limitations make this task difficult. The quantity of cancer DNA in plasma is limited, especially at early stages; this could limit sensitivity[22]. Typical ctDNA mutation screening methods can be error prone, leading to reduced specificity; the evolutionary nature of cancer also translates to an exorbitant amount of possible mutations to be screened to achieve a consistent biomarker[23]. While the use of 5-Methylcytosine as a biomarker can address some of these concerns due to its higher consistency in cancer samples[24], the bisulfite conversion process used to interrogate DNA methylation damages DNA[25]. An interrogation method with a high molecular conversion rate and a consistent set of cancer biomarkers is essential to ensure a high sensitivity[26].

Here, we describe PanSeer, a blood-based cancer screening test, and demonstrate the early detection of cancer using a unique set of samples collected as part of the Taizhou Longitudinal Study (TZL)[27]. PanSeer interrogates cancer-specific methylation signatures, and demonstrates the early detection of multiple cancer types up to four years prior to conventional diagnosis in a large-scale retrospective longitudinal study.

## Results

**PanSeer assay development.** We defined a set of differentially methylated CpG sites using publicly available microarray and Whole Genome Bisulfite Sequencing (WGBS) data from The Cancer Genome Atlas (TCGA)[28] and genomic regions known to be cancer-related in the literature[29–35], as well as internal Reduced Representation Bisulfite Sequencing (RRBS) data from a variety of cancer tissues. From these sources, we compiled a targeted panel of 595 genomic regions (Supplementary Data 1) for further interrogation in plasma samples.

We sought to interrogate these targets in a single assay with high accuracy and efficiency. Tumor DNA tends to be rare in plasma, especially in patients with early stage cancer; because conventional methods for sequencing library construction incorporating bisulfite conversion and double-stranded ligation typically have a high DNA loss rate[36], detection sensitivity can be limited. We therefore chose a Singlera library construction method utilizing semi-targeted PCR. Semi-targeted PCR requires only a single ligation event[37–39] and a single PCR primer per amplicon[40], allowing single-molecule counting at a higher molecular recovery rate than conventional methods; this gives the PanSeer assay the potential to achieve high sensitivity even in early-stage cancers.

Previous methylation-based detection methods have typically either targeted a small number of regions at high depth through PCR[41], or a large number of regions at low depth through whole genome bisulfite sequencing (WGBS) or RRBS[42]. More recently, techniques have been described to target a large number of regions at higher sequencing depth[43–45]. The PanSeer assay interrogates 595 regions at high depth; this reduces the effects of patient variability or target dropout. To demonstrate the robustness of the PanSeer assay, we performed limit of detection studies by spiking fragmented cancer cell line DNA (HT-29) into pooled healthy plasma samples. We demonstrated that the PanSeer assay can detect spike-ins down to a cancer DNA fraction of 0.1% (see "Methods" section, Supplementary Fig. 1); as demonstrated by previous studies, a high cancer detection power can be achieved with a combinatorial modeling approach[19].

**PanSeer marker identification and annotation.** In order to further identify a set of informative genomic targets that could differentiate cancer tissue from healthy tissue, we acquired a set of 200 DNA samples isolated from fresh cancer and healthy tissue from BioChain, a commercial biospecimens provider. We processed these samples using the PanSeer assay and identified a set of 477 differentially methylated regions (DMRs, see "Methods" section). In order to ensure that identified signals were originating from cancer tissue, we limited all downstream analysis of cell-free DNA samples processed by the PanSeer assay to these 477 cancer-specific DMRs (Supplementary Data 1, see "Methods" section); these regions were associated with 657 genes and 10,613 CpG sites.

We next sought to annotate the genomic regions present in the PanSeer assay that could discriminate between healthy tissue and cancer tissue. As expected, many well-known cancer-related genes or gene families were utilized by the PanSeer classifier, including FOX family genes[46], HOX family genes[47], NKL family genes[48], PAX family genes[49], and TBX family genes[50]. Some genes have been utilized previously for non-invasive cancer diagnosis in plasma, such as SEPT9 and SHOX2[25,51]. Analysis of GO terms[52] for the PanSeer genes showed that many genes were associated with DNA binding or transcription factor activity, which implies that the methylation state of these genes may contribute to gene expression changes associated with cancer. In line with expectations based on the panel design, we did not observe any major difference in gene representation across different cancer tissue types, as the targets present in the PanSeer assay had been previously selected to represent a set of ubiquitously aberrantly methylated cancer genes.

**Study design.** As part of the TZL study[27], 123,115 healthy subjects aged 25 to 90 years provided blood samples for long-term storage from 2007 to 2014; these individuals were then indefinitely monitored for cancer occurrence through linkages with local cancer registries and health insurance databases. By the end of 2017, a total of 575 initially healthy subjects (who presented as asymptomatic) were diagnosed with one of five common cancer types (stomach, esophagus, colorectum, lung or liver) within 4 years of initial blood draw (Fig. 1). These five cancer types were chosen because they had high incidence rates in the Taizhou cohort and in combination account for the highest mortality in

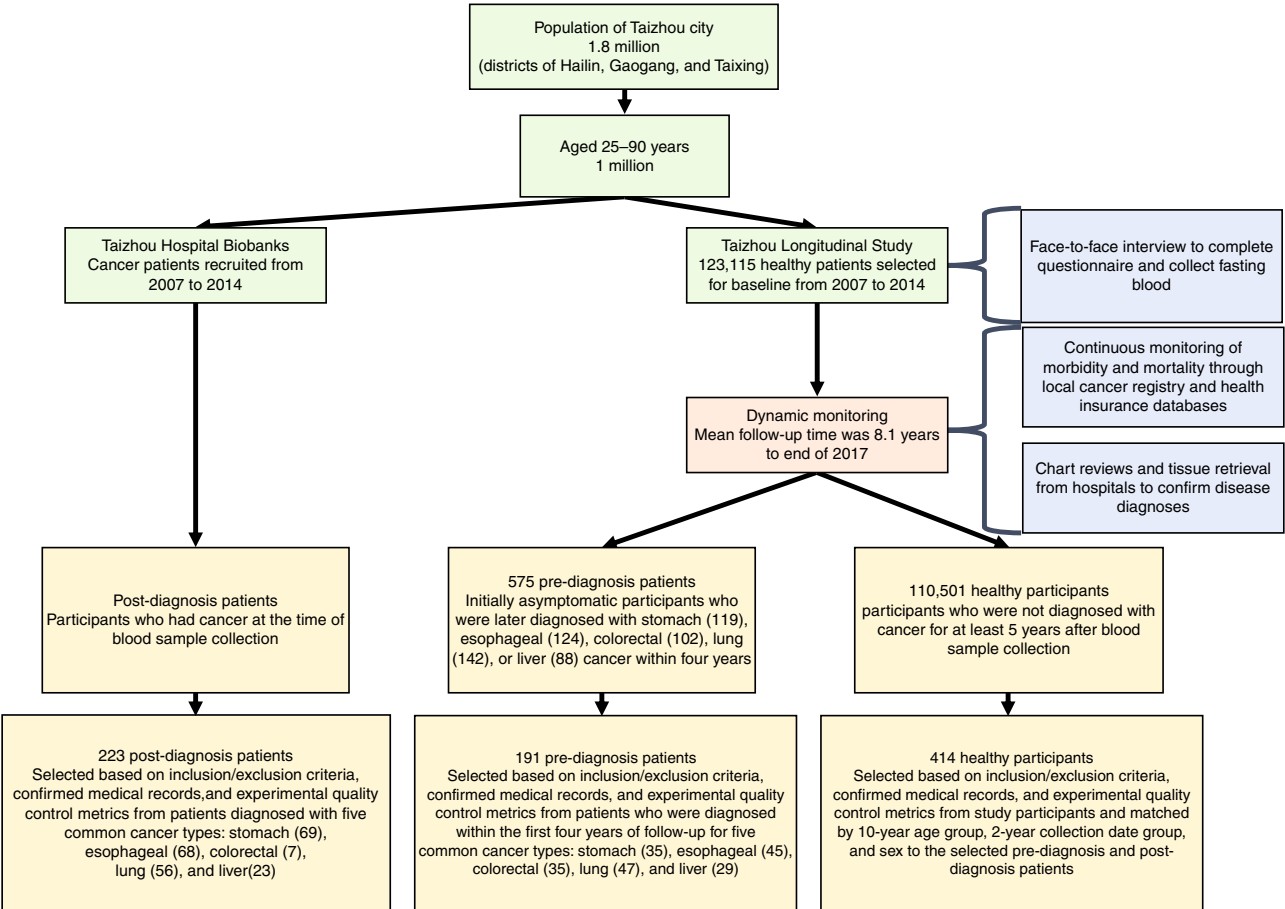

**Fig. 1 Summary of the Taizhou longitudinal study (TZL).** The flowchart shows recruitment, baseline survey, sample collection, and cohort follow-up for TZL. Qualified pre-diagnosis patients and healthy participants were selected from the TZL cohort and qualified post-diagnosis patients were selected from local Taizhou hospital biobanks; 328 samples were processed but later excluded due to not meeting inclusion criteria or failing quality control metrics.

China[53]. By retrospectively interrogating the initially collected blood samples, we were able to assess if cancer could be identified prior to conventional diagnostic methods.

We first selected 221 pre-diagnosis samples out of 575 initially asymptomatic patients who were later diagnosed with stomach, esophagus, colorectal, lung, or liver cancer within 4 years that met inclusion criteria, and 221 healthy samples out of 110,501 healthy participants not diagnosed with cancer for at least 5 years, matched on a one-to-one basis by 10-year age group with similar age distribution (Supplementary Fig. 2), sex, and collection date based on inclusion criteria (see "Methods" section, Supplementary Table 1). We then collected 357 post-diagnosis samples and 357 healthy samples matched on a one-to-one basis by 10-year age group, sex, and collection date to the post-diagnosis samples based on inclusion criteria (see "Methods" section, Supplementary Table 1). Samples with a low number of uniquely mapping DNA molecules were removed (as well as their matched healthy or cancer sample), leaving 191 pre-diagnosis samples, 223 post-diagnosis samples, and 414 healthy samples (see "Methods" section). Median patient age was 62 (ages 35–85) (Supplementary Data 2). Samples were randomly split into a training set for ensemble model development and an independent leave-out test set for model validation at an equal ratio using a random number generator. The test set samples were set aside until model development was completed and model parameters were locked down using only the training set samples.

**Development of a logistic regression classifier.** The PanSeer assay interrogates 11,787 CpG sites across 595 regions in the genome[34] using a median of 12 ng of plasma DNA, approximately 2 million sequencing reads, and a minimum of 200,000 mapped unique DNA molecules. We first computed the average methylation fraction (AMF) across each targeted genomic region for each sample (see "Methods" section). We then developed a machine learning method to classify samples as being derived from healthy patients or patients with cancer (see "Methods" section). We utilized the 207 healthy, 110 post-diagnosis samples, and 93 pre-diagnosis samples from the training set, and trained an ensemble logistic regression (LR) classifier using the AMF values for these samples; we utilized an established method[54] for this process in order to avoid overfitting (Supplementary Note 1). Training set samples were randomly split into two groups: one for model fitting and one for model validation. The model fitting set was utilized to train an LR classifier, and model scores were computed for the model validation set. This process was repeated for 1000 different random splits of the training set, and model scores were averaged across each sample to produce the final results; results for each individual sample split are shown in Supplementary Data 4 and Supplementary Fig. 3. The final classifier is therefore an average of 1000 LR classifiers built on different splits of the training set. This classifier achieved 88% sensitivity for post-diagnosis samples and 91% sensitivity for pre-diagnosis samples at a chosen specificity of 95% in the training set (Table 1, Supplementary Data 2). After evaluation in the training

**Table 1 Accuracy of PanSeer.**

| Category | Total | Training set | | | Test set | | |
|---|---|---|---|---|---|---|---|
| | | # of Samples | Specificity (%, 95% CI) | Sensitivity (%, 95% CI) | # of Samples | Specificity (%, 95% CI) | Sensitivity (%, 95% CI) |
| Healthy | 414 | 207 | 94.7 (90.7–97.3) | | 207 | 96.1 (92.5–98.3) | |
| Post-diagnosis | 223 | 110 | | 88.2 (80.6–93.6) | 113 | | 87.6 (80.1–93.1) |
| Pre-diagnosis | 191 | 93 | | 91.4 (83.8–96.2) | 98 | | 94.9 (88.5–98.3) |
| 0–1 year before diagnosis | | 22 | | 100 (84.6–100) | 21 | | 95.2 (76.2–99.9) |
| 1–2 year before diagnosis | | 21 | | 90.5 (69.6–98.8) | 23 | | 95.7 (78.1–99.9) |
| 2–3 year before diagnosis | | 19 | | 94.7 (74.0–99.9) | 31 | | 93.6 (78.6–99.2) |
| 3–4 year before diagnosis | | 31 | | 83.9 (66.3–94.6) | 23 | | 95.7 (78.1–99.9) |

Sensitivity and specificity for the training set and test set are presented and divided into subcategories by the number of years prior to cancer diagnosis by conventional testing.

set, model parameters were frozen and unchanged for all downstream analysis. We additionally demonstrated that alternative machine learning methods provided similar accuracy metrics (Supplementary Note 2), that model performance remained identical even if post-diagnosis samples were excluded (Supplementary Note 3), that model performance remained high even with a minimum read depth requirement (Supplementary Note 5), with stricter marker selection criteria based on cancer tissue (Supplementary Note 6, see below).

**PanSeer accurately detects cancer in post diagnosis samples.** We applied the LR classifier to the 113 post-diagnosis and 207 healthy samples in the leave-out test set. The receiver operating characteristic (ROC) curve for the test set is shown in Fig. 2a. The overall sensitivity of the classifier was 88% in the post-diagnosis cancer patients (Table 1, Fig. 2b, Supplementary Fig. 6) with a specificity of 96% (Table 1, Fig. 2b, Supplementary Fig. 6). Sensitivity was similar for early-stage and late-stage cancer samples (Fig. 2c, Supplementary Fig. 4), and ranged from 75% in colorectal cancers to 96% in lung cancers (Fig. 2d, Supplementary Fig. 5). While the false positives likely represent misclassifications by the PanSeer assay, it is possible some patients might have undetected cancer that was not yet diagnosed over the duration of the TZL study; we took a conservative approach and classified these samples as false positives.

**PanSeer detects cancer up to four years before diagnosis.** An important feature of a screening assay is to diagnose cancer earlier than conventional methods[21]. We therefore evaluated the ability of the ensemble LR classifier to detect cancer in the 143 apparently healthy patients that were later diagnosed with cancer within 4 years. In these pre-diagnosis cancer patients, we observed an overall sensitivity of 95% in the leave-out test set (Table 1, Fig. 2b, Supplementary Fig. 6). Sensitivity was similar for patients that were eventually diagnosed with early-stage and late-stage cancer (Fig. 2e), and ranged from 91% in esophageal cancer to 100% in liver cancer (Fig. 2f). Sensitivity appeared to be similar between patients diagnosed one to four years later, regardless of cancer stage at conventional diagnosis (Supplementary Fig. 7).

We further performed a covariate analysis using the Kruskal-Wallis test (with post-hoc testing) on patient age, patient sex, collection date, collection site, smoking status, non-cancer disease status, number of unique DNA molecules observed, and total DNA quantity extracted to ensure that other clinical or collection factors were not contributing to the PanSeer assay's ability to

detect cancer before conventional diagnosis (Supplementary Figs. 8–18). While variability was observed between different sample subsets, due to the low incidence rate of cancer, it was difficult to draw specific relationships between covariates and assay accuracy. After post-hoc testing, the only relationship observed with an effect on assay accuracy was a reduction in specificity for healthy samples collected before 2010 (representing approximately one quarter of the healthy samples in this study), while sensitivity remained consistent (Supplementary Fig. 10); it is possible that the longer storage time for these samples may have caused a reduced specificity due to DNA degradation (Supplementary Fig. 10). Despite this variability, overall sensitivity and specificity remained high regardless of clinical covariate status; the PanSeer assay was able to utilize methylation signals to detect cancer up to four years before conventional diagnosis regardless of clinical covariates.

Due to the retrospective nature of the TZL, we could not be certain whether the pre-diagnosis samples classified as normal were due to misclassification by the PanSeer assay or if these patients truly were cancer-free at the time of sample collection and developed cancer entirely after blood sample collection; we took a conservative approach and classified these samples as false negatives.

**Methylation differences between tissue and plasma samples.** While the PanSeer assay showed high sensitivity for cancer detection in both post-diagnosis and pre-diagnosis plasma samples, we observed that some genomic loci did not show consistent methylation changes between cancer tissue and cancer plasma samples. These genomic regions either showed hypermethylation in cancer tissue and hypomethylation in cancer plasma, or vise versa. While this is likely due to either intrinsic differences in tissue and plasma methylation, the variability of cancer methylation patterns, or the inherent variation in sampling methylation patterns in cell-free DNA, the possibility exists that these discordant patterns may indicate the presence of an unknown confounding factor. We therefore conducted an additional analysis of the PanSeer data in which we further filtered out any genomic regions not showing concordant hypermethylation/hypomethylation between cancer tissue and training set cancer plasma samples (Fig. 3, Supplementary Note 6). We determined that 277 genomic regions showed concordant methylation between tissue and plasma; when using only these 277 genomic regions for modeling, leave-out set sensitivity (85.0% for post-diagnosis samples, 89.8% for pre-diagnosis samples) and specificity (95.1% for healthy samples) remained high (Fig. 3,

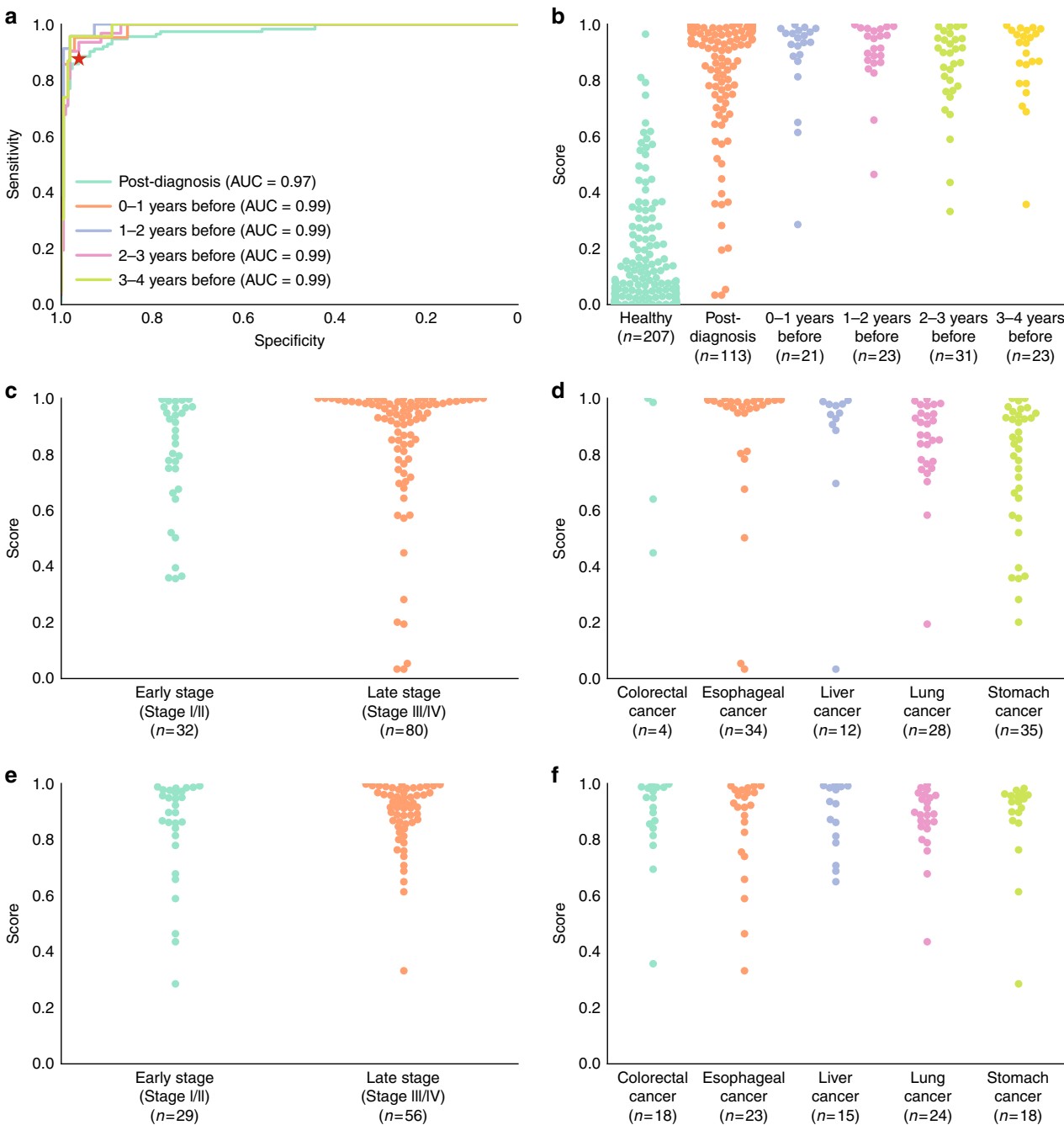

**Fig. 2 Performance of PanSeer.** All presented results used only the test set samples. Dots represent the logistic regression (LR) score. **a** Receiver operator characteristic curves (ROC) and area under the curve (AUC) values for PanSeer. The red star shows the cutoff value derived from the training set. Separate curves are shown for post-diagnosis samples and pre-diagnosis samples (divided by years before diagnosis). **b** LR scores for PanSeer samples by years before diagnosis. **c** LR scores for PanSeer samples by cancer stage for post-diagnosis samples. **d** LR scores for PanSeer samples by tissue of origin for post-diagnosis samples. **e** LR scores for PanSeer samples by cancer stage at diagnosis for pre-diagnosis samples. **f** LR scores for PanSeer samples by tissue of origin for pre-diagnosis samples.

Supplementary Note 6). While this demonstrates that the PanSeer assay can detect early-stage cancer even with a more strictly chosen target set, further large-scale longitudinal studies should be conducted to confirm early detection of cancer in pre-diagnosis samples.

## Discussion

In summary, we demonstrated that five types of cancer can be detected through a DNA methylation-based blood test up to four years before conventional diagnosis. The PanSeer assay utilizes

methylation biomarkers to their fullest extent by sensitively targeting 10,613 CpG sites across 477 genomic regions and utilizing a machine-learned ensemble score based on hundreds of genomic regions simultaneously.

The PanSeer assay was able to successfully detect five cancer types using a common set of methylation markers regardless of tissue-of-origin. As such, the genes included in the LR classifier represent a core epigenetic signature common to multiple cancer types. These genes may merit further investigation in a therapeutic context, as a change in epigenetic regulation of these

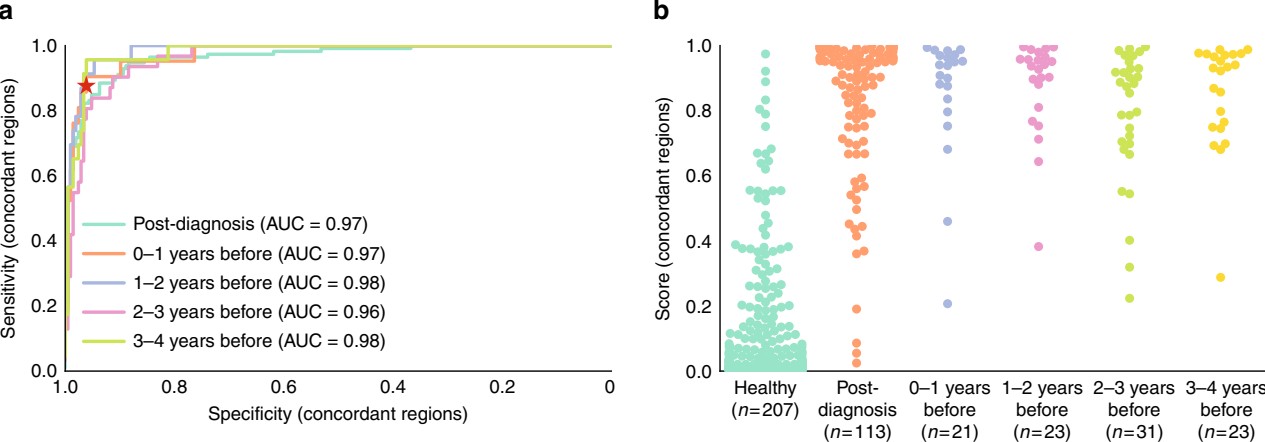

**Fig. 3 Performance of PanSeer using only tissue-concordant genomic regions.** All presented results used only the test set samples, and only utilized target regions showing concordant hyper/hypo-methylation between training set cancer plasma samples and cancer tissue samples. Dots represent the logistic regression (LR) score. **a** Receiver operator characteristic curves (ROC) and area under the curve (AUC) values. The red star shows the cutoff value derived from the training set. Separate curves are shown for post-diagnosis samples and pre-diagnosis samples (divided by years before diagnosis). **b** LR scores by years before diagnosis.

genes seems to be a common cancer phenomenon. While we have demonstrated early detection of cancer four years before conventional diagnosis through use of a longitudinal cohort, we would like to emphasize that the PanSeer assay is likely not predicting patients that will later develop cancer. Instead, the assay is most likely identifying patients who already have cancerous growths but who remain asymptomatic to current detection methods and standard of care, as many cancers do not cause the appearance of symptoms until late in disease development[21].

The possibility of a blood-based early cancer screening test has recently been investigated through multiple approaches[30,55,56], with a recent consensus forming around the high utility of cell-free DNA methylation as a cancer marker[57,58]. When developing a screening test aimed at either a high-risk or average risk population, cost is a critical factor to ensure test availability and adoption. While some previous studies have demonstrated that DNA methylation can be utilized to non-invasively both detect and determine tissue-of-origin of a cancer[45,59], these studies required the use of a large number of tissue-specific markers and a high amount of input DNA such that more than one blood vial would be required, which would incur a higher testing cost. The PanSeer assay was solely developed to detect cancer regardless of the tissue-of-origin by targeting a limited number of genomic regions that are commonly aberrantly methylated across different cancer types, allowing it to be used as a potential first-line inexpensive cancer screen; it also requires a comparatively small amount of input DNA (from only a single tube of blood). We therefore envision a clinical context where PanSeer could be used as a first-line screen; any patient testing positive on PanSeer would then undergo a more expensive reflex blood test and/or follow-up imaging to allow tissue of origin mapping. Pathological examination could then confirm the presence of cancer.

Several limitations of our study should be acknowledged. First, while this study was longitudinal, analysis was retrospective and included a matched proportion of cancer and healthy samples in order to allow development of an accurate cancer detection model; whether the PanSeer assay would improve patient outcomes still remains to be established and would require a longitudinal prospective study. Second, due to the TZL timeframe[27], modern plasma preservation techniques were not used during sample collection, leading to various degrees of genomic DNA contamination and in some cases a high sample failure rate; additionally, only 1 mL of plasma was available for each sample

(Supplementary Data 2). It is likely that with the 10 mL blood draws typically used in current protocols and better plasma preservation techniques, more and higher quality DNA molecules would be available for the assay and detection sensitivity could be further increased. Third, the spectrum of cancers observed in the TZL cohort (Fig. 1) did not exactly match the Chinese general population[53]; these differences could be due to local factors (such as lifestyle, pollution, or genetic composition). Fourth, the total number of pre-diagnosis cancer samples in this study is limited; this limitation is unavoidable, as the incidence rate of cancer in a healthy population is low. Fifth, as mentioned above, the PanSeer assay was solely developed to detect cancer regardless of the tissue-of-origin; a greatly expanded panel incorporating a large number of tissue-specific markers would need to be utilized in order to allow tissue of origin mapping. Sixth, due to the longitudinal nature of the TZL and the consent given by patients, we were unable to obtain tissue samples from the pre-diagnosis patients after they were later diagnosed; therefore, in order to ensure that identified signals were derived from cancer tissue, we utilized 200 primary tumor and normal tissue samples from a commercial biospecimens provider to choose our markers. Seventh, due to the longitudinal nature and design of the TZL, we were unable to obtain the full cancer stage information for all pre-diagnosis patients, as the TZL database internally only tracked if a patient was diagnosed at early (I/II) or late (III/IV) stage. We attempted to obtain stage information for as many patients as possible, and have described why stage information was unavailable for each patient missing this data in Supplementary Data 2; the most common reason for missing stage information was that the patient died from cancer shortly after diagnosis without surgery. Finally, as discussed above, some genomic regions did not show consistent methylation patterns between cancer tissue and cancer plasma; while this may be due to intrinsic differences between the sample types, it is possible that some unknown confounding factor may be present. We conducted further analysis to remove as much uncertainty as possible (Supplementary Note 6), but additional validation of pre-diagnosis cancer detection in a large longitudinal study is necessary to fully confirm these results.

The PanSeer assay provides a preliminary demonstration of early detection of multiple cancer types four years prior to conventional diagnosis in a robust manner, and lays the foundation for a non-invasive blood test for early detection of cancer in a

high-risk (or average-risk in the future) population. While much of current cancer research is focused on developing new therapeutics, studies have shown that early detection has the potential to reduce both treatment cost and mortality rates from cancer by a significant amount[21]. The five cancer types studied here account for 261,530 yearly cancer deaths in the US[4] and 2.1 million yearly cancer deaths in China[53]; early detection could greatly reduce deaths from these diseases. Recent studies have also identified that early detection of cancer could reduce cancer treatment costs by $26 billion annually (and also reduce the loss of productivity caused by cancer)[60]. In the future, to fully establish the clinical utility of PanSeer and fully validate the results of pre-diagnostic detection of cancer, we hope to proceed with a large prospective study of healthy individuals to determine if non-invasive cancer screening can reduce cancer deaths in a cost-effective manner.

## Methods

**Study design**. The Taizhou Longitudinal Study (TZL) started in July 2007, with a goal to recruit 200,000 people in the city of Taizhou, Jiangsu province, China, and follow the participants for at least 40 years[27]. Taizhou sits at the center of China, downstream of the Yangtze River. While the regional population and economy are average for China, the incidence of digestive cancers in Taizhou is high, at the levels of 36.9, 29.8, and 30.0 per 100,000 person-years for esophageal, gastric, and liver cancer, respectively[61]. According to the surveillance of cancer deaths in 2010, the cancer mortality was 154.05/100,000 person-years, nearly twice the mean incidence rate for China[62].

The baseline survey of the Taizhou Longitudinal Study took place during 2007–2016 in the Taixing, Gaogang, and Hailing areas in Taizhou. All men and women aged 30–75 who were living in these districts were eligible. Potentially eligible citizens were delivered an invitation letter by local community leaders and health workers, with support from the government for extensive publicity campaigns and health promotions. A Regional Coordinating Centre (RCC) and survey teams were set up for the baseline survey, consisting of 40 full-time staff members with medical qualifications and fieldwork experience in the three studied districts. All participants were indefinitely monitored for cancer occurrence through linkages with local cancer registries and health insurance databases. Exposure data were collected through survey questionnaires and physical measurements. In addition, blood samples and other biological samples were collected. A total of 123,115 individuals have been recruited and the average follow-up time is 8.1 years.

Using samples from the Taizhou Longitudinal Study (TZL), we set out to develop a classification model that could identify cancer in a non-invasive manner prior to the appearance of cancer symptoms and conventional cancer diagnosis. The study was approved by the Human Ethics Committee of Fudan University. Written informed consent was obtained from all study participants prior to inclusion in the TZL. For healthy samples, it was required that the individual was not diagnosed with cancer for the duration of the monitoring period (minimum of 5 years). For pre-diagnosis samples, it was required that a positive diagnosis of lung, liver, stomach, esophageal, or colorectal cancer was determined within 4 years of initial blood draw. For post-diagnosis samples, it was required that a positive diagnosis of lung, liver, stomach, esophageal, or colorectal cancer was determined prior to initial blood draw, and that patients were treatment naïve. Exclusion criteria were incomplete clinical information, plasma volume less than 1 mL, or evidence of hemolysis in plasma. For all samples, after processing and sequencing, it was required that at least 200,000 unique mapped DNA molecules were observed in the sequencing data, as lower amounts indicate a low-quality sample.

The study statistical plan incorporated group sizes of 144 control individuals and 144 case patients; this sample size was sufficient[63,64] to verify that the PanSeer assay had an expected sensitivity and specificity of 75% with a power of $1-\beta = 90\%$ and a significance level of $\alpha = 0.05$. Additional case and control patients were incorporated into the study due to their availability.

A total of 1156 plasma samples were collected from the TZL cohort and local Taizhou hospital biobanks for inclusion in this study. In the TZL cohort, a total of 575 initially healthy study participants were later diagnosed with colorectal, esophageal, liver, lung, or stomach cancer within 4 years. Out of these 575 pre-diagnosis samples, 191 samples were selected that passed the study's inclusion/exclusion criteria, had confirmed cancer diagnosis by retrospective chart review of hospitalization records and biopsy pathology, and passed experimental quality control metrics. 191 healthy samples passing the study's inclusion/exclusion criteria and matched by time of collection, sex, 10-year age group, and unique mapped DNA molecule count to the pre-diagnosis samples were then randomly selected from the 110,501 TZL healthy samples not diagnosed with cancer for at least five years (Supplementary Table 1). 223 post-diagnosis plasma samples passing the study's inclusion/exclusion criteria were then collected from local Taizhou hospital biobanks; 223 healthy samples passing the study's inclusion/exclusion criteria and matched by time of collection, sex, 10-year age group, and unique mapped DNA

molecule count to the post-diagnosis samples were then randomly selected from the 110,501 TZL healthy samples not diagnosed with cancer for at least 5 years.

Samples were randomly split into a leave-in training set and a leave-out test set for data analysis at a ratio of approximately 50%:50% (using a random number generator). For the training set, 110 post diagnosis cancer samples, 93 pre-diagnosis samples, and 207 healthy samples (matched by sex, 10-year age group, collection date, and unique mapped read count) were selected; these samples were used for model building, and model parameters were fixed prior to any analysis of test set samples. For the matched leave-out test set, 113 post-diagnosis, 98 pre-diagnosis samples, and 207 matched healthy samples were selected. For the leave-out test set, clinical outcomes were concealed from the classifier until calls were made.

**Tissue sample collection**. Extracted DNA from 160 cancer and 40 healthy tissue samples was purchased from Biochain (D8235086-1, D8235090-1, D8235152-1, D8235248-1). DNA was fragmented using Covaris shearing to a mean size of 150 bp to mimic the size of cell-free DNA from plasma. DNA was then end-repaired (New England Biolabs, E6050L).

**Plasma sample collection**. Blood samples were collected from 123,115 individuals from 2007 to 2014 and separated into plasma as part of the Taizhou Lontiduinal Study (TZL)[27]. Ten milliliter of blood were drawn from each patient into a K2 EDTA vacutainer and stored at 4 °C until the end of the business day. Blood samples were then centrifuged, and plasma was aliquoted into a barcoded cryovial for long-term storage at −80 °C or below. 578 healthy plasma samples and 221 pre-diagnosis plasma samples were obtained from the TZL biobank based on inclusion and exclusion criteria. Patient age at the time of blood sample collection, patient sex, cancer diagnosis date, and cancer tissue of origin was cataloged for each sample (Supplementary Data 2). 357 post-diagnosis plasma samples collected and stored with the same protocol and timeframe as the TZL samples were also obtained from local Taizhou hospital biobanks based on inclusion and exclusion criteria, and were processed in the same manner.

Cell free DNA (cfDNA) was extracted from 1 mL plasma using the QIAamp Circulating Nucleic Acid kit (Qiagen, 55114) and eluted into 50 μL of buffer according to the manufacturer's instructions with the exception of a 1 h incubation period at 60 °C during the lysis step. Carrier RNA was used to improve recovery rate (as per the manufacturer's suggestions). Samples were quantified using the Qubit fluorometric method (using 2 μL of volume); we would like to note that it is possible that carrier RNA could affect the observed DNA yield, but would bias all quantifications equally.

**PanSeer assay**. The tissue DNA and cfDNA was bisulfited converted using the Methylcode Bisulfite Conversion Kit (ThermoFisher, MECOV50) according to the manufacturer's protocol; 30 μL of the remaining 48 μL were used for each sample, as this is the maximum volume allowed by the Methylcode kit. Converted DNA samples were then processed into sequencing libraries at Singlera Genomics and sequenced on an Illumina NextSeq 500 in paired end 300 bp mode. Briefly, the bisulfite converted DNA was dephosphorylated and ligated to a universal adapter with a unique molecular identifier (UMI). Following a second strand synthesis and purification, the DNA underwent a semi-targeted PCR to target 595 genomic regions covering 11,787 CpG sites. Following a purification, a second PCR added sample specific barcodes and full length Illumina sequencing adapters. The libraries were then quantified using the KAPA Library Quantification Kit for Illumina (KK4844) and sequenced on an Illumina NextSeq 500 in paired-end 300 bp mode aiming for approximately 2 million reads per sample.

Initial data analysis was performed as part of the standard Singlera methylation sequencing processing pipeline. Reads were demultiplexed using the Illumina bcl2fastq software v2.20.0.422 (https://support.illumina.com/sequencing/sequencing_software/bcl2fastq-conversion-software.html). For each sample, paired-end read FASTQ files were merged into single reads using PEAR v0.9.6 (https://sco.h-its.org/exelixis/web/software/pear/doc.html). Reads in the merged FASTQ file were then adapter-trimmed using trim_galore v0.4.0 (https://www.bioinformatics.babraham.ac.uk/projects/trim_galore/). The unique molecular identifier sequence for each read was then added to the read name using UMI_tools v0.5.5 (https://github.com/CGATOxford/UMI-tools). The trimmed reads were then aligned to the bisulfite-converted human reference genome (version hg19) using Bismark v0.17.0 (https://www.bioinformatics.babraham.ac.uk/projects/bismark/) and Bowtie2 v2.3.1 (http://bowtie-bio.sourceforge.net/bowtie2/index.shtml). Plasma samples with less than 200,000 uniquely mapped DNA molecules were excluded from downstream processing due to low quality libraries; if a sample was removed, its matched normal/cancer sample was also removed to maintain a balanced sample set. This left a matched set of 191 pre-diagnosis samples, 223 post-diagnosis samples, and 414 healthy samples, as well as 200 healthy and cancer tissue samples.

For each remaining sample, aligned reads were assigned to each of the 595 target regions covered by the PanSeer assay based on mapped genomic position. The average methylation fraction (AMF) was computed for each target region by summing the number of observed cytosines at all covered CpG sites and dividing by the total sequencing depth at all covered CpG sites in each region per the

formula:

$$\frac{\sum_i^M N_{C,i}}{\sum_i^M \left( N_{C,i} + N_{T,i} \right)} \qquad (1)$$

where $I$ represents a CpG site in this target region, $M$ is the total number of CpG sites in this target region, $N_{T,I}$ represents the number of thymines observed at CpG site $I$, $N_{C,I}$ represents the number of cytosines observed at CpG site $i$. A graphical example of this computation is provided in Supplementary Fig. 19. This process resulted in a matrix with 595 rows (one for each target region) and 1020 columns (one for each sample); these AMF values have been provided as Supplementary Data 5.

**Marker selection**. In order to ensure that target regions utilized for cell-free DNA analysis showed aberrant methylation in cancer tissue, we utilized the set of 160 cancer tissue samples and 40 healthy tissue samples (including 8 primary lymphocyte samples) obtained from Biochain. For each of the 595 target regions, we performed a $t$-test with Benjamini-Hochberg multiple testing correction comparing the AMF values for cancer tissue samples to the AMF values for healthy tissue samples for each tissue type. 477 genomic regions showing statistically significantly different AMF values (with corrected $p$-value cutoff α ≤ 0.05) in one or more tissue types were retained for downstream processing; all other positions were removed from the AMF matrices. These resultant 477 genomic regions corresponded to 10,613 CpG sites that could differentiate the Biochain healthy lymphocyte and healthy tissue samples from the Biochain cancer tissue samples (Supplementary Fig. 20). In order to additionally confirm that the chosen sites truly represented a pan-cancer signature, we verified that PanSeer target regions showed differential methylation between healthy and cancer tissue for other tissue types in the publicly available TCGA dataset[28] (Supplementary Fig. 21).

**Cancer detection algorithm**. In order to classify each plasma sample as being derived from a healthy patient or cancer patient, a logistic regression (LR) classifier was constructed using the training set samples; in order to avoid overfitting, a cross-validation approach was utilized[54]. The detailed procedure used to build the classifier is described below, and code (with inline pseudocode in the comments) is provided in Supplementary Note 1:

1. The AMF matrix was first subset to contain only the columns corresponding to training set samples; leave-out test set samples were not considered until model parameters were finalized.
2. In order to ensure that genomic regions considered by the model were covered across all samples, any regions for which any sample did not contain at least one read were dropped from consideration in the model (i.e., any row in the AMF matrix with a missing value was removed from the matrix). 471 out of 477 genomic regions remained after this filtering step.
3. A target array with length equal to the number of training set samples was defined. If a training set sample was healthy, its corresponding target array value was set to 0, while if a training set sample was cancerous, its corresponding target array value was set to 1.
4. The training set was repeatedly randomly split at a 50%:50% ratio into a model-building set and a model validation set. This splitting process was repeated 1000 times, leading to 1000 different model-building and model validation sets (all subsets of the training set). This cross-validation procedure allows estimation of model parameters using only half of the training set and validation of model parameters using the other half of the training set.
5. For each model-building set, the scikit-learn (http://scikit-learn.org) package's LogisticRegression module was used to machine-learn parameters for an LR classifier using the AMF matrix as training data and the target array as the target values. The liblinear solver implemented in scikit-learn (http://www.csie.ntu.edu.tw/~cjlin/liblinear) was utilized for this process, and a LASSO penalty was utilized in order to build a robust model; the scikit-learn package's LogisticRegressionCV function was utilized to choose the LASSO penalty parameter using only the model-building samples. This resulted in 1,000 different LR equations (one for each model-building set), which were used as an ensemble.
6. These equations were stored. Each equation was of the form:

$$P = 1 - \frac{1}{1 + e^{-\left[ \left( \sum_{i=1}^n M_i X_i \right) + B \right]}} \qquad (2)$$

where $P$ is the probability that a given sample is cancerous, $n$ is the number of genomic regions covered by the PanSeer assay, $I$ is the current haplotype, $X_i$ is the AMF value of the $i$th genomic region, $M_i$ represents a linear coefficient fit by the LR machine learning module, and $B$ represents an intercept coefficient fit by the LR machine learning module.

1. These equations were then used to compute LR scores for each corresponding model validation set; this resulted in a matrix of LR scores,

with each training set sample having multiple scores from all iterations where it was part of the model validation set.
2. A final LR score was computed for each training set sample by averaging all scores computed when a sample was part of the model validation set. A cutoff of 0.583 was chosen based on the final training set scores to achieve a specificity as close to 95% as possible; samples with scores above this value were considered cancerous, while samples with scores below this value were considered healthy. Model accuracy was computed for the training set using these average scores and cutoffs (Table 1). This ensemble model consisting of the average result of 1000 equations was then locked down prior to any analysis of the leave-out test set.
3. For all leave-out test set samples, LR scores were computed using all 1000 equations, and a final score was computed by averaging these individual LR scores. The final score was compared to the training set cutoff, and model accuracy was computed for the leave-out test set using these average scores (Table 1).

By utilizing this ensemble classifier approach, the risk of overfitting is greatly reduced (as it can be ensured that training set and test set performance is similar). The full equations for the ensemble Logistic Regression classifier (including coefficients, intercepts, and cutoffs) is presented in Supplementary Data 3, and source code has been included in Supplementary Note 1. In addition, this same procedure was repeated using a Linear Discriminant Analysis classifier instead of Logistic Regression in order to demonstrate that results are independent of the chosen machine learning method; source code and results are presented in Supplementary Note 2. To further demonstrate the robustness of this approach, we additionally determined that the logistic regression score was unaffected by either the number of missing values in a sample or its bisulfite conversion rate (Supplementary Figs. 22, 23). We also repeated logistic regression modeling with two additional constraints to confirm that observed methylation signals in pre-diagnosis and post-diagnosis patients were cancer-derived; even with a minimum read depth requirement (Supplementary Note 5) or using stricter marker selection criteria based on cancer tissue (Supplementary Note 6), model performance remained high.

**Statistical analysis**. Means and standard deviations or medians and range were utilized to summarize continuous variables, while whole numbers and percentages were utilized to summarize categorical variables. Accuracy metrics were computed for each sample set and subset based on sample covariates. Sensitivity is defined as true positives/(true positives + false negatives). Specificity is defined as true negatives/(true negatives + false positives). Binomial confidence intervals for sensitivity and specificity were calculated using the Clopper-Pearson method. To determine if any sample covariates impacted assay performance, the Kruskal-Wallis $H$-test was utilized to compare model scores for each category (healthy, pre-diagnosis, and post-diagnosis) across each analyzed covariate (Supplementary Figs. 8–18); if necessary, the Mann–Whitney $U$-test was utilized to perform post-hoc testing.

**Limit of detection study**. In order to measure the analytical limit of detection of the PanSeer assay, a set of spike-in samples consisting of a mixture of cancer cell line DNA and healthy plasma was constructed; it was determined whether each spike-in level could be separated from baseline healthy plasma. To create the spike-in samples, DNA from the HT-29 cell line was purchased from ATCC and sheared to approximately 150 bp (Covaris). The DNA was then purified and concentrated using Ampure beads. Plasma from multiple healthy individuals was pooled together to use as a baseline in the limit of detection study. Sheared HT-29 was spiked into the pooled plasma at molar ratios of 0% (baseline), 0.1, 0.5, 1.0, 5.0, and 10%, with six technical replicates for each spike-in ratio.

The PanSeer assay was then run on the spike-in samples. In order to evaluate the limit of detection analytically, due to the variation in methylation levels across the genome, four baseline samples were chosen as training samples to determine the level of observable background methylation in healthy plasma across each genomic region. For each individual genomic region, a cutoff value was determined using these four baseline training samples; this cutoff was set at three standard deviations above the mean methylation value observed in the baseline samples. For the remaining two 0% spike-in samples and all 0.1–10% spike-in samples, the number of genomic regions for which each sample was above the computed cutoff value was totaled and plotted in Supplementary Fig. 1; all replicates for all spike-in samples could be distinguished from the baseline samples, with higher spike-in molar ratios resulting in more genomic regions above the baseline value. Detailed code for the limit of detection analysis is provided in Supplementary Note 4.

**Reporting summary**. Further information on research design is available in the Nature Research Reporting Summary linked to this article.

## Data availability

Data from this study, including the methylation matrices, are available in the main text, supplementary materials, supplementary datasets, or have been deposited in GitHub in the repository NCOMMS-20-10056-T [https://github.com/ncomms-20-10056-t/ncomms-20-10056-t]. Full genetic sequencing data was not included in the informed

consent, hence only the methylation status at each genomic position has been released. The TCGA dataset is available at at the GDC Data Portal [https://portal.gdc.cancer.gov/].

## Code availability

The Python code utilized in this study has been deposited in GitHub in the repository NCOMMS-20-10056-T [https://github.com/ncomms-20-10056-t/ncomms-20-10056-t].

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

## Acknowledgements

We are grateful to the staffs at Fudan University Taizhou Institute of Health Sciences. We also appreciate the members of the survey teams and the participants for their contribution to this study. The TZL study was supported by the National Key Research and Development Program of China (grant number: 2017YFC0907000, 2017YFC0907500, 2019YFC1315800, 2019YF101103, and 2016YFC0901403), the National Natural Science Foundation of China (grant number: 91846302 and 81502870), the Key Basic Research grants from the Science and Technology Commission of Shanghai Municipality (grant number: 16JC1400501), the International S&T Cooperation Program of China (grant number: 2015DFE32790), the Shanghai Municipal Science and Technology Major Project program (grant number: 2017SHZDZX01), the International Science and Technology Cooperation Program of China (grant number: 2015DFE32790), and the 111 Project (B13016). Funding for the DNA methylation assays was provided by Singlera Genomics.

## Author contributions

X.C., J.G., A.G., Y.G., K.Z., R.L., and L.J. designed the study, wrote and revised the manuscript. X.C., M.L., W.Y., S.Y., and L.J. supervised the TZL study. X.Y., Y.J., T.Z., C.S., X.Z., M.F., X.W., Y.Y., J.Z., Z.Y., and J.W. collected and organized samples from the TZL cohort. R.L., J.G., Q.H., X.L., L.C., Z.Z., H.N., Z.L., Z.X., H.S., J.D., and C.M. performed or supervised experimental protocols. A.G. and J.M. analyzed the experimental data and developed the classification model. All authors approved the final manuscript.

## Competing interests

J.G., A.G., Q.H., J.M., X.L., L.C., Z.Z., H.N., Z.L., Z.X., H.S., J.D., C.M., and R.L. are employees of Singlera Genomics. Y.G. and R.L. are board members of Singlera Genomics. J.G., A.G., and R.L. are inventors on a patent (US62/657,544) held by Singlera Genomics that covers basic aspects of the library preparation method used in this paper. K.Z. is a co-founder, equity holder, and paid consultant of Singlera Genomics. The terms of these arrangements are being managed by the University of California–San Diego in accordance with its conflict of interest policies. X.C., M.L., Z.Y., X.Y., Y.J., T.Z., C.S., X.Z., M.F., X.W., Y.Y., J.Z., J.W., S.Y., W.Y., and L.J. declare no competing interests.
