## [Peer Review File · Nature Communications]

This manuscript has been previously reviewed at another journal that is not operating a transparent peer review scheme. This document only contains reviewer comments and rebuttal letters for versions considered at Nature Communications.

Reviewers' Comments:

Reviewer #1:

Remarks to the Author:

I thank the authors for addressing many of the main concerns I had, and I do think that this revised version is overall an improvement. Unfortunately, however, some of the bigger concerns I had remain:

1) Fitting of LR-model: Although the authors have now tried to do a nested CV, there are no sufficient details provided in Methods to check that the authors have implemented this correctly. For instance, I did check Suppl.Note.1, and there is a line of code which states `LogisticRegressionCV(penalty='l1', solver='liblinear', random_state=random_state, Cs=[1.0, 5.0, 10.0, 50.0, 100.0], cv=3)`. It would appear that the authors may have run LR with a Lasso-penalty(?), yet this was never mentioned in the whole manuscript. And does `cv=3` mean that they split the training set into 3 portions? This was also never mentioned in the whole manuscript. And what does the “Cs” parameter refer to- I presume this is the penalty parameter? This was also not mentioned in the Methods section of the paper. In other words, what is terribly unclear and not explained at all in the whole manuscript, are details concerning the specific process of parameter estimation, optimization and performance evaluation in training and the leave-in test sets. I suggest that the authors provide in the manuscript, detailed pseudocode (not Python or R or any other programming-specific code), so that anyone can check over the logical implementation of the model fitting and evaluation procedure in all sets including the leave-out test set. Indeed, it is quite amazing that in this revised version the performance measure in the leave-out test set has shot up to e.g. a whopping 96% sensitivity 3-4 years prediagnosis. In the original version of the paper the authors reported 100% sensitivity in training set and 60% sensitivity in test set for the 3-4 years prediagnosis setting, which indicated clear overfitting. Now, the authors claim near 100% sensitivity in both sets, independently of pre or postdiagnosis, with even better performance in prediagnostic samples compared to postdiagnosis (!), and independently of the time between sample collection and diagnosis. I find this all very counterintuitive.

We appreciate the reviewer's suggestion to utilize a nested CV approach with a larger training set in their previous comments on the original manuscript, as it has greatly improved the results presented in our revision. As part of the revised manuscript, we provided a full Python notebook containing both dependencies, code, and model results; we had thought that this notebook would be self-explanatory. We did utilize a LASSO procedure in order to help build a robust model, but omitted this from the methods section. We also utilized the scikit-learn package's `LogisticRegressionCV` function in order to perform Logistic Regression penalty parameter estimation in an unbiased manner; this function estimates the optimal LASSO penalty using the training set. We have added additional descriptive text to both the Methods section and the Supplementary Note to further explain these function parameters. These factors are essentially related to a preference on presentation style, not a weakness or lack of rigor for the analysis. Furthermore, we would like to emphasize that in Supplementary Note 2, we demonstrated that 1

second method also led to consistent results, hence the manuscript conclusions are robust regardless of choice of modeling method.

2) Treatment of 3 groups in LR-classifier construction: it would now appear that the authors treated the prediagnostic samples on an equal footing to the cancer ones (as per their explanation in their rebuttal letter), however in the main text at the bottom of page-4, the authors continue to call prediagnostic samples as “patients with cancer”, when at the time of sample-draw the cancer was not present or asymptomatic. Could the authors please do an analysis comparing only healthy to prediagnostic samples (i.e. not including postdiagnostic cases). It would be interesting and important to see the performance measures for a classifier not built at all with postdiagnosis samples. I note that I had indirectly asked for this in my previous review.

In both the original manuscript and this draft, we put the pre-diagnostic samples on equal footing with the post-diagnostic samples, as we do not believe that the PanSeer assay is identifying cancer before it is present; we feel that pre-diagnostic samples are likely existing cancers that are still asymptomatic. We state in the Discussion that “we would like to emphasize that the PanSeer assay is likely not predicting patients that will later develop cancer. Instead, the assay is most likely identifying patients who already have cancerous growths but who remain asymptomatic to current detection methods and standard of care.”

We thank this reviewer for suggesting a comparison between only healthy and pre-diagnosis samples. We have attached the results below for the leave-out sample set of building a model utilizing only the healthy and pre-diagnostic samples; the model calls were identical whether or not the post-diagnosis samples were included in training. We have included this analysis and the supporting figures/tables in Supplementary Note 3, and have also included the results below.

	Original Model	Pre-Diagnosis Only
Pre-Diagnosis Sensitivity	94.9%	94.9%
Specificity	96.1%	96.1%

3) No benchmarking to number of NAs/amount of cfDNA: I am glad that the authors agree that replacing NAs with 0s is not sensible. However, the authors reported in the original version that

the number of NAs was in effect an indirect measure of the amount of cfDNA in the sample. This then raised the concern as to whether the predictive performance was independent of the number of NAs (since the authors had replaced NAs with 0s). In my previous review I therefore also asked the authors to demonstrate that their DNAm-based classifier is a better diagnostic and prediagnostic tool than the amount of cfDNA in the sample. In other words, what the authors have not shown is that their DNAm based classifier is a better diagnostic and prediagnostic classifier than the number of NAs in the sample (surrogate for amount of cfDNA). That their classifier is now derived from a data matrix that does not contain NAs is a completely separate issue, which does not fully address my main point.

In order to address the reviewer's previously stated concern about NAs potentially influencing the model result, we removed all genomic regions with any missing values from the data matrix. By doing this, we unequivocally demonstrated in our revised manuscript that the cancer-derived methylation signal in plasma is present in the methylation data itself and is not an artifact caused by missing values. To address the reviewer's remaining concern about missing values potentially being a competitive diagnostic metric, we performed a comparison between the logistic regression scores and the number of missing values in the data matrix:

Note that the correlation coefficient between the logistic regression score and the number of missing values is only 0.05, indicating that there is no correlation. This result further demonstrates that missing values do not influence the model results and that the cancer methylation signal is present in the methylation data. We have included this analysis as Supplementary Figure S22 in the revised manuscript. The data set that we generated from these samples is large and there are many potential angles to analyze the data. The objective of our study is to demonstrate robust detection of cancer signature in the pre- and post-diagnosis samples, and we clearly met this goal. The manuscript should not be criticized on other potential

analyses that can be done on this data set but were not demonstrated, because we believe those are outside of the scope.

Reviewer #2:

Remarks to the Author:

The authors have properly addressed several of my initial concerns. However some concerns still remain:

Original comment #1. Now that the authors disclosed the target regions used in their classifier, it is important to present some heatmaps with DNA methylation data from normal blood cells (WGBS data can be obtained from IHEC, blue print, roadmap, etc). This will show how much of this signal is originated also from blood cells rather than tumor cells only. Another heatmap of all CpGs overlapping TCGA data on all cancer types and normal tissue will show what cancer types do and do not have DNA methylation on these coordinates and what normal tissues could confound the results. Since all these data-sets are publicly available, it should not be a problem to present these results and would show a more clear picture of the regions being used and limitations of selecting the targets upfront in this type of study.

In order to demonstrate that our target regions show signal in tumor cells and not blood cells, we processed 8 commercially purchased primary lymphocyte samples using the PanSeer assay, and have included a heatmap comparing these lymphocytes to healthy and cancerous colon tissue, stomach tissue, lung tissue, and breast tissue. The PanSeer target regions show differential methylation between cancer tissue samples and healthy tissue/blood samples, indicating that the signal contribution from blood cells and healthy tissue is minimal. The same heatmap also shows that all four cancer types show similar methylation patterns across the PanSeer target regions.

In order to expand this analysis, we have additionally included a heatmap showing the average methylation status of CpG sites covered by both the PanSeer target regions and the TCGA cancer and normal tissue methylation array data set using cancer types not covered in this study (cervix, cholangiocarcinoma, pancreatic, glioblastoma, head/neck squamous cell, kidney, prostate, sarcoma, skin, and uterine, respectively). As visible in the heatmap, the PanSeer target regions show at least some methylation changes across many different cancer types (though some cancers show more methylation changes than others).

We have included these heatmaps as Supplementary Figures S20 and S21 in the revised manuscript.

Original comment #4 (spike in experiments for sensitivity). I agree with the authors comments regarding the mutation detection. But still, other papers have shown higher sensitivity than 0.1% using DNA methylation (Shen et al., Nature, 2018 Fig 1B for instance). It would be good to have a similar analysis down to 0.001% using PanSeer for direct comparison.

We thank the reviewer for the comment and are glad they agree with our previous assessment of mutation detection. The ultimate goal of any technical benchmarking is to determine the performance limit in a synthetic setting which mimics the real situation as close as possible. While Shen et al 2018 did demonstrate a limit of detection for methylation lower than 0.1% with cfMeDIP-seq, a direct comparison between the limit of detection for cfMeDIP-seq and PanSeer is neither appropriate nor necessary. We would like to emphasize that the PanSeer assay is a small targeted panel (~600 targets) designed for use with low input DNA amounts (~10 ng) and a low amount of sequencing (~2M reads per sample). The cfMeDIP-seq assay, on the other hand, is a whole-methylome high depth sequencing approach which captures millions of genomic regions but requires a higher input DNA amount (60 ng for spike-in experiments) and much higher sequencing depth. Despite capturing the entire methylome with 60 ng of DNA, only two differentially methylated regions were observed in the cfMeDIP-seq spike-in experiment at 0.001%. As the PanSeer assay is designed to be a targeted panel aimed at 10-20

ng of input DNA (3000-6000 genomic copies) that matches the amount of plasma DNA available in this project, we felt that a 0.1% spike-in would better represent the limit of detection for a targeted panel with low sequencing depth and input DNA amount. Additional proof down to 0.001% would not have any impact on the conclusion, as the limiting factor is the amount of plasma DNA available.

Original comment #6 (bisulfite conversion rate). I'm especially concerned about the conversion rates presented in the rebuttal. Seems most of the samples are below 99% conversion rate, with many samples below 95% conversion. This can create a high rate of false positive DNA methylation calls (unmethylated CpG not being converted and called as methylated CpG). If the abundance of circulating tumor DNA in pre-symptomatic samples is expected to be below 0.1% and the noise is 1-5%, this is 10-50 fold higher noise than signal. This problem is compounded in the revised manuscript, now that the authors dropped the haplotype block type of analysis. The haplotype block was presented in the Guo S et al Nat Genet.2017 paper as a bioinformatics mechanism to increase robustness against these false positive methylation calls due to bisulfite conversion failures and the current type of analysis is now susceptible to this problem.

We used standard bisulfite conversation methods and a widely adopted bisulfite conversion kit. The bisulfite conversion rate should be comparable to (not any better or worse than) most studies in the literature, including several bisulfite-based qPCR or sequencing studies on cancer detection with cell-free DNA (Leontiou et al PLOS One 2015, Bock et al Nature Biotechnology 2016, Kint et al PLOS One 2018). We discovered an error in calculating the bisulfite conversation rate in our manuscript, and would like to thank this reviewer for catching that. When reporting the estimate for bisulfite conversion rate, we utilized an output from the Bismark alignment software of level of observed non-CG methylation. However, we mistakenly utilized only CHH contexts rather than CHN contexts from the output file; this led to an incorrect computation of the bisulfite conversion rate. We have corrected this computation, and have updated the bisulfite conversion rates; the rates are now higher.

While incomplete bisulfite conversion can lead to incorrect results, because the level of incomplete conversion observed here is spread evenly across all samples in the data set and multiple target regions are utilized by the model, the model is able to correctly identify samples as cancerous or healthy even with this level of incomplete conversion. It is possible that lower levels of methylation may not be detectable using the PanSeer assay due to incomplete bisulfite conversion. However, we did not see any correlation between model score and bisulfite conversion rate. We have included this figure as Supplementary Figure S23.

Reviewer #3:

Remarks to the Author:

Chen et al. have presented an improved version of their manuscript. They present PanSeer, which uses targeted bisulphite sequencing to measure DNA methylation levels in cell-free DNA. In the previous version of the manuscript, the method description was insufficient to judge the manuscript and several methodological concerns were raised. In this version of the manuscript, the methods section is much clearer, and the authors have addressed most of the concerns that were previously raised. However, I still have one concerns, which influences the application of the PanSeer classifier beyond this study.

Major concern:

1. The authors are able to build a model to classify samples into those that have a cancer diagnosis and those that don't, and the model gives very good sensitivity and specificity values. However, the authors have a collection of 191 pre-diagnosis samples, 223 post-diagnosis samples and 414 healthy samples. The proportion of cancer vs healthy individuals in their data is roughly 50%. This percentage is way higher than the real prevalence of cancer in a real population. For example, in the TLS study, the authors report a prevalence of cancer is 0.5% (575 out of 123,115). In a machine learning classification problem, considering this huge class

imbalance is crucial to address. Therefore, the sensitivity and specificity of PanSeer that the authors present in their manuscript can't be translated to a real-life scenario.

This comment is on the potential impact of this study, not related to the rigor of our analysis. As we mentioned in the discussion, this was a retrospective analysis of a previously collected prospective data set, and we are therefore unable to utilize the current study to comment on clinical outcomes. For this study, we oversampled the number of cancer patients in order to ensure we were building an accurate detection model; this same practice has been utilized in previous cancer detection manuscripts during model building (Diehl et al PNAS 2005, deVos et al Clinical Chemistry 2009, Cohen et al Science 2018, Chen et al Journal of Cancer 2019). We have further emphasized this limitation of our study in the Discussion section, and have further clarified that we intend to next apply the PanSeer assay in a prospective study.

Minor concerns:

2. The input files that are used in the python code in the author's repository should be made available as well. For example, the file "tsh_amf_20191016.tsv" is not part of the Github repository.

We thank the reviewer for this comment. This is a typographical error, as the "tsh_amf_20191016.tsv" file should actually be "DataS1.tsv"; an incorrect filename was mistakenly present in the code. We have updated the code on GitHub to fix the typo.

Reviewers' Comments:

Reviewer #1:

Remarks to the Author:

I have gone through the manuscript again in some detail and have even looked at some of the data provided with the submission. I am now satisfied that the implementation of the machine learning methods is OK. After all, inspection of the data as provided in SuppDataS1 reveals that over 62% of the 447 markers have a $P < 0.05$ between healthy and pre/postdiag cases, with 36% passing Bonferroni significance ($P < 0.05/447$), and approximately 100 markers passing an even stringer $1e-6$ threshold. So, this strong signal means that prediction accuracies would be high. However, I am afraid that two major puzzles remain, which I consider major concerns: First, the authors report very high accuracy when comparing healthy to only prediagnostic cases, and in fact, the whole paper indicates that classification accuracy is comparable between prediagnostic and postdiagnostic cases, if not even better for prediagnostic ones. Given that the total amount of DNA extracted was much higher for the cancer samples, and that the amount of DNA used for library construction was much lower for prediagnostic ones (this follows from the data presented in Table S3: the AUC comparing healthy to prediagnostic cases in terms of DNA input is 0.82), I find it hard to reconcile these numbers with the better classification performance on prediagnostic cases, unless of course there is another source for the DNA methylation differences in the prediagnostic ones, a source that is unrelated to cancer. I strongly suspect this is what is happening because if the authors were to explore the DNAm profiles for some of the top-ranked regions, they will see that the difference in DNA methylation is much wider between healthy and prediagnostic cases than between healthy and postdiagnostic ones, i.e. the pattern is non-monotonic. This is counterintuitive and makes no sense if the marker is really coming from the tumor DNA. In other words, the discrimination accuracy between healthy and cancer as reported in this manuscript is probably correct, but I fear that the reported discrimination accuracies between healthy and prediagnostic cases is inflated due to some confounder, which would therefore render the conclusions of this paper invalid.

The second concern which may shed light on the first problem above relates to my previous 3rd concern that the authors had not shown that their diagnostic classifier does better than the amount of cfDNA. The scatterplot figure shown in their rebuttal, which plots the number of missing values against the LR scores does not really address my point for various reasons: first of all this figure does not label in color the samples as to whether they are healthy, prediagnostic or postdiagnostic. However, perhaps more worryingly, when I glance at SuppFig.S19, and I see the definition of the AMF-values, I don't think it is at all statistically justified to include calls based on only 1 read. Some reads have all sites methylated, other reads have them unmethylated, so the error could be very high if we only use 1 read, right? There could be many values in the AMF-matrix which are based on only 1-5 reads, and so without proper quantification of the uncertainty associated with these values, their data matrix may be confounded. For instance, could it be that for the prediagnostic cases for which the DNA input was so much lower than for healthy and postdiagnostic cases, that for these prediagn samples the number of AMF-values derived from say only 1-3 reads is much higher? The authors have not adjusted for this and other technical variation in the data-matrix, prior to implementation of the machine learning method. I realise that the authors have provided many SuppFigs. displaying performance measures as a function of various technical measures, but this type of "a-posteriori" analysis is not rigorous and is subject to easy misinterpretation. In order to make progress here and hopefully resolve the paradox in this paper, I would suggest to perform a very standard SVD. Do an SVD on the row-centered AMF-matrix (ie the one used prior to running the LR-classifier). What do the top components correlate with? Does it correlate with the mean number of uniquely mapped reads per region? If it does, as I think it does, then the authors should have adjusted for this before running any machine learning method. I think that proper adjustment for technical variation present in the AMF-matrix would probably lead to a lower prediction performance for the prediagnostic samples, which would make a lot more sense and bring the results in line with biological intuition.

Another major concern which I raise now because this was raised by another reviewer, is that there is in fact only a very moderate correlation (Pearson ~ 0.31) between the t-statistics of differential methylation for the plasma samples (comparing healthy to pre+posdiag) and the t-statistics derived from the BioChain tissue samples (comparing normal to cancer). In fact, I note that the great majority of the 477 loci exhibit hypermethylation in cancer based on the Biochain tissue samples, but that this skew towards hypermethylation is *****NOT***** observed in the plasma samples. In fact, the top-ranked loci according to the AMF matrix in plasma exhibit equal numbers of hypo and hypermethylation, which suggests to me that many of the markers in the LR-classifier are not derived from tumor DNA. Instead of heatmaps which are not a good way to show *****quantitative***** data (heatmaps are only good for qualitative views) I would suggest that the authors generate scatterplots of t-statistics between healthy and pre+postdiag plasma samples, vs the corresponding t-statistics in the BioChain samples. The authors should also do this for each cancer-type separately, and then also separately comparing healthy to prediagn and healthy to posdiagnosis when subsequently correlating these t-statistics to those from the Biochain normal-cancer comparison. I think that these scatterplots would exhibit much stronger correlations for the postdiagnostic samples and also when stratifying by cancer-type, compared to the correlations when using prediagnostic samples. These analysis may well support the view that in the case of the prediagnostic samples, many of the DMR loci do not derive from tumor-DNA.

Reviewer #2:

Remarks to the Author:

The authors have addressed my previous comments.

I would recommend the authors to revise the citations to improve scholarship. Important papers in the cfDNA methylation field addressing very similar questions were not cited nor discussed in the discussion section. The last manuscript below uses a very similar approach to the one presented by the authors. I would recommend the authors to have a paragraph explaining the state of the art in the field, the advantages of their approach and the limitations (inability to discriminate between cancer types?) compared to the previous literature.

-CancerDetector: Ultrasensitive and non-invasive cancer detection at the resolution of individual reads using cell-free DNA methylation sequencing data
Nucleic Acid Research, 2018 Sep;46(15):e89.

- Sensitive tumour detection and classification using plasma cell-free DNA methylomes. Nature
2018, 563 (7732), 579-583

- Sensitive and specific multi-cancer detection and localization using methylation signatures in cell-free DNA. <https://doi.org/10.1016/j.annonc.2020.02.011>

Reviewer #1 (Remarks to the Author):

I have gone through the manuscript again in some detail and have even looked at some of the data provided with the submission. I am now satisfied that the implementation of the machine learning methods is OK. After all, inspection of the data as provided in SuppDataS1 reveals that over 62% of the 447 markers have a $P < 0.05$ between healthy and pre/postdiag cases, with 36% passing Bonferroni significance ($P < 0.05/447$), and approximately 100 markers passing an even stringer $1e-6$ threshold. So, this strong signal means that prediction accuracies would be high.

We thank the reviewer for taking the time to thoroughly review our manuscript and supplementary data. We greatly appreciated the reviewer's suggestions, and feel that the suggested improvements to our machine learning approach and additional analyses have further ensured that the conclusions of this study are robust.

However, I am afraid that two major puzzles remain, which I consider major concerns: First, the authors report very high accuracy when comparing healthy to only prediagnostic cases, and in fact, the whole paper indicates that classification accuracy is comparable between prediagnostic and postdiagnostic cases, if not even better for prediagnostic ones. Given that the total amount of DNA extracted was much higher for the cancer samples, and that the amount of DNA used for library construction was much lower for prediagnostic ones (this follows from the data presented in Table S3: the AUC comparing healthy to prediagnostic cases in terms of DNA input is 0.82),

We thank the reviewer for these comments. In order to address these concerns, we performed a more detailed analysis of the DNA quantities in Table S3.

We first inspected the total amount of DNA extracted from each plasma sample. With the exception of a few outlier samples, we found that the total amount of DNA extracted was similar across both healthy and cancer samples (see plot below). This result matched our expectations, as while late stage cancer patients are known to show higher levels of circulating DNA in plasma, there are many factors that can affect the total amount of DNA present that are unrelated to disease state.

We next looked at the DNA quantity used for library construction. We were very surprised by the reviewer's comment, as in our experimental SOP, we loaded the same volume for each sample. Our DNA extraction kits elute into 50 uL of volume; we utilized 2 uL for quantification, and then loaded 30 uL of extracted DNA into the assay (up to a maximum of 20 ng); therefore, most samples should have 62.5% (or less) of the original DNA quantity loaded into the experiment.

When we observed that Table S3 did not match expectations from our experimental protocol, we went back to our original laboratory notes and sample tracking files, and discovered that an error in an Excel VLookup formula had caused incorrect DNA quantities to be listed in Table S3 for many samples. We apologize for the error, and thank the reviewer for catching this mistake! We have updated Table S3 with the correct DNA quantities; with the corrected numbers, the amount of DNA loaded for library construction is much more similar. We have also added a clarification about our DNA loading strategy to the Methods section of the manuscript. In order to further confirm that our sample sheet did not contain any mixups, we checked the ratio of chrX and chrY reads in the sequencing data for each sample; we correctly called the sex of each patient sample.

Some post-diagnosis samples had a higher DNA quantity utilized due to the higher overall DNA yield from these samples. However, DNA quantity did not correlate with the logistic regression model score (see below).

I find it hard to reconcile these numbers with the better classification performance on prediagnostic cases, unless of course there is another source for the DNA methylation differences in the prediagnostic ones, a source that is unrelated to cancer. I strongly suspect this is what is happening because if the authors were to explore the DNAm profiles for some of the top-ranked regions, they will see that the difference in DNA methylation is much wider between healthy and prediagnostic cases than between healthy and postdiagnostic ones, i.e the pattern is non-monotonic. This is counterintuitive and makes no sense if the marker is really coming from the tumor DNA. In other words, the discrimination accuracy between healthy and cancer as reported in this manuscript is probably correct, but I fear that the reported discrimination accuracies between healthy and prediagnostic cases is inflated due to some confounder, which would therefore render the conclusions of this paper invalid.

We appreciate the reviewer bringing up the concern that there may be a confounding factor leading to high discrimination accuracy in our data set. This concern is the most important factor to address. We were careful during study design to match clinical covariates such as age, sex, and smoking status in order to reduce the amount of potentially confounding factors. We have addressed the potential confounding variable of DNA input concentration above. We also appreciate the reviewer listing specific analyses that could ensure that no confounding factors exist in his second and third comments; we have addressed these concerns more specifically in response to these comments below. **Taken together, we feel that our analyses have demonstrated that the observed methylation signals in cancer plasma are derived from circulating cell-free tumor DNA, rendering our conclusions valid.**

We would also like to comment on the monotonicity of methylation data in cancer. While many biomarkers show a monotonic pattern linked to disease state, this is not necessarily true when it comes to cancer methylation. The Cancer Genome Atlas (TCGA) study demonstrated that methylation patterns in cancer can be highly variable within cancer types, stages, and even within different samples from the same tumor. Recent studies have even demonstrated that

variability in methylation patterns can itself be utilized as a cancer biomarker (Hansen et al, Nature Genetics 2011). When looking to perform non-invasive detection, cancer methylation signals can become even more variable due to the use of cell-free DNA. We would therefore not necessarily expect that observed cancer methylation signals in our assay would behave in a monotonic manner.

Nonetheless, in order to directly address the reviewer's concern, we have plotted the methylation pattern for the top markers in our study in the figure below. We observed that in most cases the post-diagnosis samples tended to have similar or stronger methylation signal than pre-diagnosis samples when compared to healthy plasma, indicating that the signal we are observing is likely coming from the tumor. We have performed additional analyses below to further demonstrate this.

The second concern which may shed light on the first problem above relates to my previous 3rd concern that the authors had not shown that their diagnostic classifier does better than the amount of cfDNA. The scatterplot figure shown in their rebuttal, which plots the number of missing values against the LR scores does not really address my point for various reasons: first of all this figure does not label in color the samples as to whether they are healthy, prediagnostic or postdiagnostic.

We apologize for this oversight. We have modified the Supplementary Figure S22 to color the samples as healthy, pre-diagnostic, and post-diagnostic. The number of missing values does not correlate with LR score regardless of sample type.

However, perhaps more worryingly, when I glance at SuppFig.S19, and I see the definition of the AMF-values, I don't think it is at all statistically justified to include calls based on only 1 read. Some reads have all sites methylated, other reads have them unmethylated, so the error could be very high if we only use 1 read, right? There could be many values in the AMF-matrix which are based on only 1-5 reads, and so without proper quantification of the uncertainty associated with these values, their data matrix may be confounded. For instance, could it be that for the prediagnostic cases for which the DNA input was so much lower than for healthy and postdiagnostic cases, that for these prediagn samples the number of AMF-values derived from

say only 1-3 reads is much higher? The authors have not adjusted for this and other technical variation in the data-matrix, prior to implementation of the machine learning method.

We thank the reviewer for this comment. While we did drop any region that had a missing value in even one sample, we did not require a minimum number of reads per region. In order to directly address this concern, we have included an additional Supplementary Note 5, in which we repeated modeling and classification but with a requirement that each region have a minimum of 10 UMIs. Both model AUC and post/pre-diagnosis sensitivity at 95% specificity (83.2% and 92.8% respectively) were not greatly affected by removing low-depth regions, indicating that read depth is not a confounding factor.

I realise that the authors have provided many SuppFigs. displaying performance measures as a function of various technical measures, but this type of “a-posteriori” analysis is not rigorous and is subject to easy misinterpretation. In order to make progress here and hopefully resolve the paradox in this paper, I would suggest to perform a very standard SVD. Do an SVD on the row-centered AMF-matrix (ie the one used prior to running the LR-classifier). What do the top components correlate with? Does it correlate with the mean number of uniquely mapped reads per region? If it does, as I think it does, then the authors should have adjusted for this before running any machine learning method. I think that proper adjustment for technical variation present in the AMF-matrix would probably lead to a lower prediction performance for the prediagnostic samples, which would make a lot more sense and bring the results in line with biological intuition.

We thank the reviewer for the suggestion. In order to address this concern, we performed a principal component analysis on the initial AMF-matrix (pre-classifier). We observed that the first principal component showed weak correlation ($r^2=0.6$) with the mean per-region read depth, and that a subset of samples showed elevated read depth and an elevated PC1. However, PC1 did not correlate with the logistic regression score, indicating that this component does not appear to be linked to the LR model results.

To fully address the reviewer’s concern, we removed any sample showing technical variability by excluding all samples with a mean per-region read depth above 200, and recomputed our LR

model accuracy; the classification model's performance remained very similar (post-diagnosis sensitivity of 89.3%, pre-diagnosis sensitivity of 94.6%, and specificity of 93.8%). While the results were not affected, we are still willing to exclude the samples with a high read depth from the manuscript if desired in order to eliminate any remaining concerns.

Another major concern which I raise now because this was raised by another reviewer, is that there is in fact only a very moderate correlation (Pearson ~ 0.31) between the t-statistics of differential methylation for the plasma samples (comparing healthy to pre+posdiag) and the t-statistics derived from the BioChain tissue samples (comparing normal to cancer). In fact, I note that the great majority of the 477 loci exhibit hypermethylation in cancer based on the Biochain tissue samples, but that this skew towards hypermethylation is *****NOT***** observed in the plasma samples. In fact, the top-ranked loci according to the AMF matrix in plasma exhibit equal numbers of hypo and hypermethylation, which suggests to me that many of the markers in the LR-classifier are not derived from tumor DNA.

We thank the reviewer for raising this point. While we initially used tumor and healthy tissue samples to aid in the initial marker selection, because of the differences between tissue and plasma samples, the two types may not behave in the same manner for every methylation marker. Because of this, we had utilized a training set of plasma samples to learn an LR model designed for plasma samples. However, to address the reviewer's concern, we have included Supplementary Note 6, in which we perform a stricter LR modeling procedure. In this analysis, we only included markers that behaved identically in tissue and training set plasma samples (i.e. requiring consistent hypermethylation or hypomethylation across both sample types). We observed that both model AUC and post/pre-diagnosis sensitivity at 95% specificity (85.0% and 90.8%, respectively) were not greatly affected by limiting to only consistent input markers, indicating that even when limited to markers that we can be almost certain are derived from tumor DNA, we can accurately classify samples as healthy or cancer; this gives us extra confidence in the PanSeer assay.

Instead of heatmaps which are not a good way to show **quantitative** data (heatmaps are only good for qualitative views) I would suggest that the authors generate scatterplots of t-statistics between healthy and pre+postdiag plasma samples, vs the corresponding t-statistics in the BioChain samples. The authors should also do this for each cancer-type separately, and then also separately comparing healthy to prediagn and healthy to postdiagnosis when subsequently correlating these t-statistics to those from the Biochain normal-cancer comparison. I think that these scatterplots would exhibit much stronger correlations for the postdiagnostic samples and also when stratifying by cancer-type, compared to the correlations when using prediagnostic samples. These analysis may well support the view that in the case of the prediagnostic samples, many of the DMR loci do not derive from tumor-DNA.

We thank the reviewer for this suggestion. In Supplementary Note 6, we have included a figure showing the t-statistics of each marker in the tissue and plasma samples. However, due to the differences in both scale of methylation changes (i.e. difference of means) and variability of methylation changes (i.e. standard deviation), the t-statistics did not show a very clear pattern. However, we found that even when limiting markers to the two consistent quadrants (i.e. markers that showed consistent hypermethylation/hypomethylation in both tissue and plasma), **we were still able to accurately classify samples as cancer or healthy.**

In order to further confirm that observed signals in plasma samples were derived from CRC tissue samples, we attempted a “negative control analysis” to build a classification model on the same data set, using only genomic regions that **did not** show significant difference between the Biochain healthy and cancer tissue samples. While all loci interrogated by the PanSeer assay had been previously shown to have links to cancer (Supplementary Table 1), we hypothesized that if our classification model was utilizing signals derived from cancer tissue, assay performance should decrease if we limited modeling to loci that did not show strong cancer signal in the Biochain tissue samples. When utilizing loci not showing cancer signal in the Biochain samples, we determined that both post-diagnosis and pre-diagnosis sensitivity greatly fell to only 45% and 46% respectively; this drop in performance indicates that the PanSeer

classification model appears to rely on tissue-derived signal. **With these additional analyses, we feel confident that we have demonstrated that by using DMR loci derived from tumor DNA, we can detect cancer in both pre-diagnosis and post-diagnosis plasma samples.**

Reviewer #2 (Remarks to the Author):

The authors have addressed my previous comments. I would recommend the authors to revise the citations to improve scholarship. Important papers in the cfDNA methylation field addressing very similar questions were not cited nor discussed in the discussion section. The last manuscript below uses a very similar approach to the one presented by the authors. I would recommend the authors to have a paragraph explaining the state of the art in the field, the advantages of their approach and the limitations (inability to discriminate between cancer types?) compared to the previous literature.

-CancerDetector: Ultrasensitive and non-invasive cancer detection at the resolution of individual reads using cell-free DNA methylation sequencing data
Nucleic Acid Research, 2018 Sep;46(15):e89.

- Sensitive tumour detection and classification using plasma cell-free DNA methylomes. Nature 2018, 563 (7732), 579-583

- Sensitive and specific multi-cancer detection and localization using methylation signatures in cell-free DNA. <https://doi.org/10.1016/j.annonc.2020.02.011>

We thank the reviewer for this suggestion. We have cited these additional manuscripts, and have added additional comments to the Discussion section about the state of the art in the field and how our approach differs from other techniques.

Reviewers' Comments:

Reviewer #1:

Remarks to the Author:

The authors have responded satisfactorily to my concerns, except for the concern regarding the substantial inconsistency in directionality observed between normal/cancer tissue and the corresponding changes in prediagn. plasma, as is clear from the scatterplots at the very end of Suppl.Note.6. The authors respond by citing "variability in cancer" as a common phenomenon and as an explanation for the observed inconsistency, but this in my opinion wrong. The Hansen et al paper showed that specific loci exhibit high variability in DNA methylation in cancer, yet the fact remains that these loci are always higher or lower methylated in cancer compared to normal i.e. you don't get loci that are hypermethylated in some cancers and hypomethylated in other cancers (relative to same normal tissue), unless the loci are hemimethylated in normal tissue. However, the great majority of the loci undergoing differential methylation in cancer are either unmethylated or fully methylated in normal tissue, and therefore changes in cancer are unidirectional. It is the level or degree of methylation change that is variable between cancer samples of the same cancer-type, not the direction! Thus, if we now assess the plasma results, it is conceivable (but unlikely) that a marker that can discriminate prediagnostic cases may fail to discriminate postdiagnostic ones simply because of a sampling issue and the variability in cancer noted above, however, what we are seeing in this study is many loci that are *significantly* hypomethylated in the plasma of prediagnostic cases, which are *significantly* hypermethylated in the cancer tissue samples and hypermethylated in the plasma of postdiagnostic cases. Given that the overwhelming majority of loci in the PanSeer assay are not mapping to hemimethylated sites in normal tissue, this leads to the conclusion that the observed hypomethylation in plasma must be caused by a source other than the tumor DNA it is trying to diagnose.

A potential but unlikely explanation here is that the whole analysis in this paper is confounded by cancer-type, and indeed I note that the classification was performed by pooling cancer-types together, which means that in principle a locus that is hypermethylated in one cancer-type (say colon) could well be hypomethylated in another (e.g. breast). However, my understanding based on the PanSeer-assay is that this is not the case and that most of the loci in the PanSeer assay show the same unidirectional increase in DNAm in cancer, regardless of cancer type. I further note that the results shown in Fig.2D & 2F, which display the performance stratified by cancer-type is open to misinterpretation, because these figures are NOT based on separate classifiers derived for each cancer-type, but simply reflect a retrospective stratification of the classification derived from the classifier trained on all cancer-types.

So, in my humble opinion, a serious question mark remains about the reported high accuracy for the prediagnostic samples, and my instinct tells me that the derived classifier will not be reproducible in independent prediagnostic cohorts. Therefore, my strong recommendation is that the authors should be open about this major limitation (the current paper completely ignores this major limitation, "sweeping it under the carpet"). The authors should state and disclose this limitation in the (i) abstract, (ii) results and (iii) discussion/conclusion sections, and should include the scatterplots in SuppNote.6 as a main figure in the paper (Suppl.Fig also OK, but this paper I note has only a few display items and there should be space for this very important and critical inclusion). The authors should further acknowledge that there is an inconsistency here, and that resolving this issue needs further investigation in future studies. Therefore, the authors need to acknowledge that the prediagnostic results are only preliminary, should be interpreted with great caution and that they require further validation.

Reviewer Response:

Reviewer #1 (Remarks to the Author):

The authors have responded satisfactorily to my concerns, except for the concern regarding the substantial inconsistency in directionality observed between normal/cancer tissue and the corresponding changes in prediagn. plasma, as is clear from the scatterplots at the very end of Suppl.Note.6. The authors respond by citing “variability in cancer” as a common phenomenon and as an explanation for the observed inconsistency, but this in my opinion wrong. The Hansen et al paper showed that specific loci exhibit high variability in DNA methylation in cancer, yet the fact remains that these loci are always higher or lower methylated in cancer compared to normal i.e. you don’t get loci that are hypermethylated in some cancers and hypomethylated in other cancers (relative to same normal tissue), unless the loci are hemimethylated in normal tissue. However, the great majority of the loci undergoing differential methylation in cancer are either unmethylated or fully methylated in normal tissue, and therefore changes in cancer are unidirectional. It is the level or degree of methylation change that is variable between cancer samples of the same cancer-type, not the direction! Thus, if we now assess the plasma results, it is conceivable (but unlikely) that a marker that can discriminate prediagnostic cases may fail to discriminate postdiagnostic ones simply because of a sampling issue and the variability in cancer noted above, however, what we are seeing in this study is many loci that are *significantly* hypomethylated in the plasma of prediagnostic cases, which are *significantly* hypermethylated in the cancer tissue samples and hypermethylated in the plasma of postdiagnostic cases. Given that the overwhelming majority of loci in the PanSeer assay are not mapping to hemimethylated sites in normal tissue, this leads to the conclusion that the observed hypomethylation in plasma must be caused by a source other than the tumor DNA it is trying to diagnose.

A potential but unlikely explanation here is that the whole analysis in this paper is confounded by cancer-type, and indeed I note that the classification was performed by pooling cancer-types together, which means that in principle a locus that is hypermethylated in one cancer-type (say colon) could well be hypomethylated in another (e.g. breast). However, my understanding based on the PanSeer-assay is that this is not the case and that most of the loci in the PanSeer assay show the same unidirectional increase in DNAm in cancer, regardless of cancer type. I further note that the results shown in Fig.2D & 2F, which display the performance stratified by cancer-type is open to misinterpretation, because these figures are NOT based on separate classifiers derived for each cancer-type, but simply reflect a retrospective stratification of the classification derived from the classifier trained on all cancer-types.

So, in my humble opinion, a serious question mark remains about the reported high accuracy for the prediagnostic samples, and my instinct tells me that the derived classifier will not be reproducible in independent prediagnostic cohorts. Therefore, my strong recommendation is that the authors should be open about this major limitation (the current paper completely ignores this major limitation, “sweeping it under the carpet”). The authors should state and disclose this limitation in the (i) abstract, (ii) results and (iii) discussion/conclusion sections, and should include the scatterplots in SuppNote.6 as a main figure in the paper (Suppl.Fig also OK, but this paper I note has only a few display items and there should be space for this very important and critical inclusion). The authors should further acknowledge that there is an inconsistency here,

and that resolving this issue needs further investigation in future studies. Therefore, the authors need to acknowledge that the prediagnostic results are only preliminary, should be interpreted with great caution and that they require further validation.

We thank the reviewer for their deep analysis of our manuscript data and conclusions. We agree with the reviewer that additional longitudinal studies must be conducted to further validate the conclusions of the manuscript in detection of pre-diagnosis samples. We have included figures from Supplementary Note 6 in the main text to demonstrate that some target regions behave different in tissue and plasma samples, and have stated that additional validation must be conducted.